# Major Depressive Disorder: Existing Hypotheses about Pathophysiological Mechanisms and New Genetic Findings

**DOI:** 10.3390/genes13040646

**Published:** 2022-04-06

**Authors:** Muhammad Kamran, Farhana Bibi, Asim. ur. Rehman, Derek W. Morris

**Affiliations:** 1Department of Pharmacy, Faculty of Biological Sciences, Quaid-i-Azam University, Islamabad 45320, Pakistan; kamifidi@gmail.com (M.K.); arehman@qau.edu.pk (A.u.R.); 2Centre for Neuroimaging, Cognition and Genomics (NICOG), Discipline of Biochemistry, National University of Ireland Galway, H91 CF50 Galway, Ireland; 3Department of Microbiology, Faculty of Biological Sciences, Quaid-i-Azam University, Islamabad 45320, Pakistan; farhanamalik234@yahoo.com

**Keywords:** gene, GWAS, major depressive disorder

## Abstract

Major depressive disorder (MDD) is a common mental disorder generally characterized by symptoms associated with mood, pleasure and effectiveness in daily life activities. MDD is ranked as a major contributor to worldwide disability. The complex pathogenesis of MDD is not yet understood, and this is a major cause of failure to develop new therapies and MDD recurrence. Here we summarize the literature on existing hypotheses about the pathophysiological mechanisms of MDD. We describe the different approaches undertaken to understand the molecular mechanism of MDD using genetic data. Hundreds of loci have now been identified by large genome-wide association studies (GWAS). We describe these studies and how they have provided information on the biological processes, cell types, tissues and druggable targets that are enriched for MDD risk genes. We detail our understanding of the genetic correlations and causal relationships between MDD and many psychiatric and non-psychiatric disorders and traits. We highlight the challenges associated with genetic studies, including the complexity of MDD genetics in diverse populations and the need for a study of rare variants and new studies of gene-environment interactions.

## 1. Introduction

Major depressive disorder (MDD) is a complex heterogeneous disorder and the most common psychiatric disorder [1]. It is ranked as a major contributor to worldwide disability, affecting over 300 million people, with an estimated annual prevalence of 4.4% of the total world population [2]. The World Mental Health Survey estimates a lifetime prevalence of 10–15% for MDD [3]. MDD can be chronic or recurrent in nature and is typically associated with prolonged periods of low mood and anhedonia [4]. It has significant socioeconomic consequences, including increased morbidity, disability, excess mortality, substantial economic costs and heightened risk of suicide [5]. MDD has a negative effect on an individual’s feelings, thinking and actions, causing an individual to feel sad and/or lose interest in activities that were previously enjoyable. MDD can lead to a variety of problems related to emotional and physical well-being that negatively affect an individual’s normal functioning both at work and at home [6]. In addition to characteristic mood symptoms and deviant thoughts, there are other symptoms related to cognition (mental ability to acquire knowledge through thought, experience and senses) and bodily functions (chronic fatigue, decreased interest in sex, decreased appetite, insomnia, lack of deep sleep or oversleeping). MDD is associated with many comorbid mental and physical illnesses that suggest complex underlying mechanisms. The underlying molecular mechanism(s) of MDD are still unknown.

## 2. Diagnosis and MDD Phenotypes

To date, there are no specific biomarkers that can be used to confirm the diagnosis of MDD for clinical or research purposes. A multifaceted approach is undertaken when diagnosing MDD that centres on a psychiatric evaluation using the Diagnostic and Statistical Manual of Mental Disorders (DSM-5), a set of diagnostic criteria developed and published by the American Psychiatric Association [7], or diagnostic criteria set out by the International Classification of Disease (ICD-10). In order to confirm the diagnosis of MDD, a physician may do a physical examination and ask questions about a patient’s health, as there can be underlying medical conditions that lead to MDD [8]. In the psychiatric evaluation, the patient is asked to fill out a questionnaire answering specific questions about symptoms, thoughts, feelings and behavior patterns [9]. DSM-5 defines a case of MDD as one where a person experiences a low mood or loss of interest or pleasure in earlier enjoyable daily activities for a period of at least two weeks, and the presence of a majority of specified symptoms such as disturbance in sleep pattern, eating, liveliness, attentiveness, or self-esteem. No omissions are specified for major depressive episode symptoms due to medical ailments, substance use disorders or medications [10]. Psychosis may also accompany MDD of varying severity [11].

No clear-cut distinction has been drawn to differentiate the depression phenotypes (probable depression, broad depression, recurrent depression, and psychotic depression). Depressive symptoms experienced may vary from individual to individual. The DSM-5 and ICD-10 classify depression based on the parameters experienced by the patient along the course of illness while considering the number of symptoms, the types of symptoms, the intensity of the symptoms and social, functional and occupational impairments in patients [12]. However, for research purposes, for example, for genetic studies [4,13,14], a range of methods has been reported to identify cases of depression in different cohorts. These methods of diagnosis range from self-reporting of depressive symptoms in biobanks, accessing electronic health records (EHRs) of individuals or scheduling structured interviews with individuals. As detailed later in Section 7, the genetic basis of these different phenotypic definitions is highly correlated with each other.

## 3. Available Treatment Options for MDD

Although MDD is a treatable disorder, treatments work for some individuals and not others. Various pharmacological and non-pharmacological interventions, including psychotherapy, electroconvulsive therapy (ECT) and transcranial management stimulation, are used in the management of MDD [15,16,17]. Psychotherapy (cognitive behavior therapy, interpersonal therapy and online psychological interventions) has been reported to have an impact on relieving symptoms [18] and on improving patients’ overall quality of life [19]. Therefore, many guidelines developed for the management of MDD recommend psychotherapy as a monotherapy or in combination with antidepressant medication.

The most common antidepressants approved by the US FDA for use in the management of MDD include selective serotonin reuptake inhibitors (SSRIs), serotonin-norepinephrine reuptake inhibitors (SNRIs), monoamine oxidase inhibitors (MAOIs), tricyclic antidepressants (TCAs), noradrenergic and specific serotonergic modulators, norepinephrine-dopamine reuptake inhibitors, multimodal antidepressants, serotonin modulators, MT1/MT2 agonists and 5-HT2C antagonists, serotonin reuptake inhibitor and 5-HT1A-receptor partial agonists, neurosteroids, and newer agents like non-competitive N-methyl-D-aspartate receptor antagonists (Table 1) [15,16,20]. The focus of research has been to optimize the treatment strategy for MDD. Although not used for diagnosis, individual biomarkers and clinical characteristics are used to predict the efficiency of management strategies. These strategies include the use of genetics, electrophysiology, neuroimaging, peripheral protein expression levels, neurocognitive performance, data on developmental trauma and the personality of an individual [21].

## 4. Pathophysiological Mechanisms of MDD

The characteristic symptoms of MDD that cause complexity in the treatment of the disorder lie within the domain of cognitive, emotional and physiological effects [20]. The currently available treatments, including antidepressants, are only effective for 50% of MDD patients, which reflects the fact that the diagnosis of MDD is solely dependent on behavioral symptoms. The treatment options of MDD are, therefore, not specific to the pathology of the disease [22]. Despite an abundance of research in understanding the neurophysiology and neuropsychiatry of MDD, the precise mechanism for the development of depression is still unknown due to the complex heterogeneous nature of the disorder, possibly involving multiple etiologies. It is crucial to understand the pathophysiological mechanisms of MDD to develop new therapeutic options [23].

Previous studies support the involvement of various pathophysiological mechanisms, such as the biogenic amine hypothesis, the receptor hypothesis, neurotrophic factors, cytokine theory and endocrine factors. A major challenge is that no single hypothesis explains all aspects of MDD due to the involvement of multiple interlinked disease mechanisms [24,25,26].

### 4.1. Biogenic Amine Hypothesis

The brain contains a vast number of serotonergic, dopaminergic and noradrenergic neurons. The process of memory working, behavioral regulation and mindfulness are all associated with the function of the prefrontal cortex, which is regulated by norepinephrine through noradrenergic neurons. Dopamine is linked with the modulation of reward and motivation, memory working and mindfulness, whereas serotonin is the largest cohesive system of neurotransmitters and innervates all areas of the brain [27]. It is well supported that several behavioral symptoms (fatigue, low mood, psychomotor retardation, reduced motivation and vigilance) are associated with monoaminergic systems. Several studies reported considerably lower levels of serotonin in depressed patient brain samples from suicide cases. Abnormalities in dopamine levels in the brain cause impaired motivation, aggregation and concentration, whereas reduced levels of norepinephrine (individually or coupled with dopamine) mediate the vast number of depressive symptoms such as aggression, sex, concentration, appetite, mindfulness and motivation [28,29]. The low availability of these neurotransmitters results in impaired cognitive functions and reduced neurotransmission, which consequently causes depression. Another underlying cause of depression may involve the reduced functioning of protein transporters or neurotransmitter receptors [30]. Various studies reported lower levels of monoamine neurotransmitters due to reduced numbers of transport proteins at cellular junctions in depression patients [31,32].

### 4.2. Receptor Hypothesis

The receptor hypothesis attempts to elucidate depression-related pathogenesis from multiple perspectives. There are a number of receptors, such as α-amino-3-hydroxy-5-methyl-4-isoxazolepropioni acid (AMPA), γ-aminobutyric acid (GABA), N-methyl-D-aspartate (NMDA), 5-hydroxytryptamine (5-HT), glucocorticoid and dopamine receptors, which have been extensively studied for their role in the occurrence and development of depression [25].

The **AMPA** receptors (heterotetrameric ion channels) are subtypes of glutamate receptors with an assemblage of four subunits (GluA1, GluA2, GluA3 & GluA4). A total of 80% of AMPA receptors are composed of the subunits GluA1 and GluA2, and the remaining 20% are composed of GluA3 and GluA4. These are found in postsynaptic membranes in the hippocampal neurons [33]. The GRIA family of genes that encodes these receptors lies in the 5q33 chromosome region. The onset of depression (aging relevant) is reported to be associated with SNPs in GRIA (rs4403097 & rs4302506). Various studies have shown changes in the distribution of AMPA subunits with MDD and an associated deterioration of plasticity related to the expression of AMPA receptors has suggested the need for the formulation of antidepressants which could potentially activate the AMPA receptors [34,35].

The **GABA** receptors are ion channels composed of four subunits (α, β, γ, δ) and are responsible for the transmission of an inhibitory neurotransmitter (GABA). These receptors are important targets of various drugs, including barbiturates, benzodiazepines and anesthetics. The gene *Gabra2,* which encodes the GABAA receptors (composed of 2 α1, 2 β2 and 1 γ2 subunits), is located on chromosome 4 [36]. Studies support the presence of defects in GABAergic synaptic transmissions, impaired homeostasis in the hippocampal region, and the dysfunction of glutamate-related synapses in mice samples upon GABAergic deficiency due to inhibition of genes encoding γ2 subunits, consequently resulting in depression [37].

The **NMDA** receptors are hetero-multimeric complexes composed of two GluN1 and two GluN2 subunits. These receptors are subtypes of ionic glutamate receptors generally expressed in the central nervous system, often associated with mood regulation. In MDD patients, the antagonists of NMDA receptors have shown positive therapeutic efficacy along with new synaptic connectivity and proficiency in reversing the neuronal deficiency induced by stress [38]. Studies have shown the abnormal expression of GluN2A (encoded by the *GRIN2A* gene in 16p 13.2) due to *GRIN2A* gene hyper-methylation, which resulted in increased susceptibility to depression [39]. Another study suggests the increased synthesis of synaptic proteins and relief in behavioral symptoms of depression upon knock out of the *GRIN2B* gene (located on 12p13.1), which encodes the GluN2B subunit [40]. A GWAS study has suggested that *GRIN2B* (based on SNP rs220549) is a potential candidate gene that is linked with the onset of depression in MDD patients [41].

Notably, ***5-HT*** (serotonin) is an inhibitory neurotransmitter with 14 receptor subtypes spread widely in synapses and the cerebral cortex. It has been found to be one of the key factors in the pathogenesis of MDD and mediates the regulation and efficacy of antidepressant drugs [42]. A significant association was observed between the serotonin transporter gene *SLC6A4* (genetic variants: 5-HTTLPR, rs140700, rs4251417, rs6354, rs25528, rs25531) and DNA methylation at various CpG sites, which suggests DNA methylation-associated depression status in a community-based population study of older individuals. Lower methylation was associated with depression in individuals homozygous for the short 5-HTTLPR and 5HTTLPR/rs25531 alleles, but only for individuals with the SS genotype of 5-HTTLPR [43]. The effect of environmental stress on gene expression and neuronal functioning through epigenetic alterations may generally represent a mechanism for disease risk. Epigenetic alterations such as changes in DNA methylation have been observed in MDD patients [44]. DNA methylation is the most-investigated epigenetic alteration in MDD patients. Generally, DNA methylation at the binding sites of enhancers results in transcriptional repression, whereas there is an antagonistic effect for DNA methylation at binding sites for repressors. Structural variations within a genomic region might also contribute to the development of the disease phenotype. For the complex structure of the *SLC6A4* gene, studies have shown that the expression of this gene is controlled by the modulation of microRNA mir-16 binding sites located in non-translating regions. miR-16 was found to be a regulator (post-transcriptional) of 5-HT levels, and expression of the *5-HT* gene was modulated by 3′-UTR microRNAs. It was observed that the reduction in the expression level of 5-HT upon overexpressing miR-16 changes the [125I]-RTI-55 and [3H]-paroxetine binding sites in the differential serotonergic cell line. An increased level of *5-HT* expression upon miR-16 reduction was also observed. Therefore, the expression of *SLC6A4* and the resultant function of 5-HTT may be strongly affected by the polymorphism within or near regions of mir-16 binding sites [45]. The first studied large-scale meta-analysis of research on depression suggested the statistically significant association of *SLC6A4* and *SLC6A3* genes among various genes of the monoaminergic system [46].

**Glucocorticoids** represent the secretion of stress-related hormones with two different receptors (low-affinity mineralocorticoid receptor (MR) and high-affinity glucocorticoid receptor (GR)) [47]. GR regulates the expression of neurotrophins that induce apoptosis in neuronal cells and alter the neurogenesis in the hippocampal region in adults [48]. The imbalance of MR and central GR causes HPA axis disorders and increased susceptibility to depression. Studies have shown affected levels of GR and HPA axis functions due to varying expression of the *NR3CI* gene that encodes GR (located on chromosome 5). Low levels of *NR3CI* expression were also found in MDD patients with suicidal attempts [49]. Epigenetic alterations have been observed in the promoter region of *NR3C1* at overlapping CpG sites in peripheral blood cells and brain tissue of patients who experienced early life trauma [50].

The **dopamine** receptors are members of the transmembrane domain G-protein-coupled receptors family, which are involved in the transmission of noradrenaline precursor (dopamine) [51]. The dopamine (D2) receptors are a key target site for antidepressant drugs, including haloperidol [52]. A polymorphism (A241G) in the gene (*DRD2*) encoding the D2 receptor, located on 5p 15.3, was observed, which has shown a correlation with early to mid-adolescence as well as maternal parenting-related depressive symptoms [53].

### 4.3. Endocrine System

Abnormalities related to the endocrine system such as alterations in growth hormone levels (GH), HPA dysfunction and abnormalities in thyroid levels have been identified as key contributors to MDD etiology [28]. The direct and indirect effects of GH on the NE system have been investigated for their contribution to the development of depression. Defective GH release was observed in depressed patients [54]. Specifically, the use of clonidine and apomorphine exhibited a blunt response of GH secretion in MDD patients [55]. The thyroid gland is responsible for producing two active forms of hormone (tri-iodothyronine T3 and tetra-iodothyronine T4) by the activation of thyroid-stimulating hormone (TSH) in the pituitary [56]. Several studies support the possible association between thyroid function alteration and depression. Thyroid hormones regulate the whole metabolic reaction in the human body. Abnormalities in thyroid functioning could possibly cause symptoms such as sleep disturbance, loss in weight, and psychomotor retardation in MDD patients [57]. Studies also suggest the indirect role of thyroid hormones as co-transmitters in the adrenergic nervous system [58].

The HPA axis is considered a significant contributor to the development of MDD in patients. In depressed patients, dysfunction in the glucocorticoid mechanism, hyper-secretion of cortisol and corticotrophin-releasing factor (CRF), the administration of exogenous glucocorticoid, reduced HPA axis suppression and impaired signaling of corticosteroid receptors have been suggested in association with alterations in the HPA axis [59]. The HPA activity is highly regulated by CRF, which increases the secretion of adrenocorticotrophic hormone (ACTH) from the interior pituitary following the stimulation of adrenal glands to secrete cortisol [60]. Several studies have reported elevated levels of CRF in the cerebrospinal fluid of depressed patients. Elevated and persistent levels of CRF are responsible for high cortisol secretion and are associated with intense and severe symptoms such as hopelessness, weight loss, disturbed sleep, psychomotor retardation and overreaction towards psychological stressors in MDD patients [59,61].

### 4.4. The Cytokine Theory

Cytokines are inflammatory chemicals secreted in response to foreign pathogenic antigens by lymphoid cells [62]. The inflammatory process in the human body, such as the regulation of monocytes, basophils, lymphocytes, neutrophils, and eosinophils, is a function of cytokines [63]. Several types of cytokines involve interleukins, C-reactive proteins, interferons, tumor necrosis factor and serum amyloid proteins [64]. The altered immune functions in MDD patients potentially indicate the role of cytokines in the development of chronic stress levels in patients. There are several studies reported in support of and in contradiction to the association of cytokines with MDD development [65]. Numerous studies have linked the inflammatory factors as a result of sickness to behavior in depression. The normal immunologic response in the human body towards pro-inflammatory cytokines and infections causes various behavioral responses, such as disturbed sleep, nausea, loss of interest and appetite, and increased body temperature, which suggest that the actions of inflammatory cytokines cause broader range of behavioral sickness responses in MDD patients [66,67].

### 4.5. Neurotrophic Factor Hypothesis

This hypothesis involves both stress hormone signaling involving cortisol and neurotrophic factor signaling, including brain-derived neurotrophic factor (BDNF) [68]. Over time, inflammatory factors in the brain and the over-excitation or over-stimulation of neuronal activity leads to a decrease in neurogenesis (formation of new neurons in the brain), contributing to the pathophysiology of MDD. Therefore, whenever there is a stressor, the brain responds to it normally by releasing cortisol [69]. The circulating levels of cortisol become higher. In a situation where the brain is unable to control or suppress the release of cortisol, it results in decreased neurogenesis. This hypothesis is relevant because most antidepressant drugs either prevent or reverse the damage caused by increased cortisol release, over-excitation, and inflammation [70,71]. Additionally, ECT is one possible adjunctive therapy for depression that canprevent or reverse the effects of cortisol and overexcitation [72]. The factor linking cortisol and stress to decreased neurogenesis is thought to be BDNF. BDNF normally stimulates dendritic sprouting and new neuron growth, but following the long-term elevation of cortisol, BDNF is decreased. Antidepressants are thought to reverse this effect by increasing BDNF levels [73]. In addition to the HPA axis [74] and the release of cortisol, additional neuroendocrine factors are at play in this hypothesis, such as steroids and endogenous steroids like the progesterone derivative allopregnanolone. The reason for the inclusion of allopregnanolone in this hypothesis is that chronic stress and depression are known to be associated with decreased allopregnanolone [75]. There is also a rapid decrease in this hormone following labor and delivery, which could contribute to the development of postpartum depression specifically [76]. Thyroid hormone is another neuroendocrine factor that can contribute to the expression of depression. The main evidence for this is that hypothyroidism or reduced thyroid hormone is a cause of secondary depression [77].

### 4.6. Neuroplasticity/Neurogenesis Hypothesis

In adults, neurogenesis is the process of the formation of new neurons and neuronal connections in a certain region of the brain (dentate gyrus of the hippocampus and lateral ventricles) [78]. According to several studies, one of the causes that describe the pathophysiology of depression is the lack of neurogenesis in adults. The disturbance in metabolic levels of neurotropic factors such as BDNF in nervous tissue is found to be the primary cause of these effects [24]. In addition, various studies have reported changes in morphological and synaptic plasticity in MDD patients [79]. In the limbic structures of adult brains, BDNF is a highly expressed neurotropic factor that maintains the survival, migration, proliferation and differentiation of neurons. The dysregulation of *BDNF* expression, such as an increase in DNA methylation of *BDNF* gene promoters, was observed in peripheral mononuclear cells of depressed patients’ blood [80]. Various studies have reported decreased levels of receptor TrkB (neurotropic receptor tyrosine kinase 2) and its genetic variant *NTRK2* in depressed and suicidal patients [81]. The antidepressant drugs work to increase BDNF levels, but the exact mechanism of how they activate *BDNF* expression is still not so clear. A recent study has shown the probable contribution of *BDNF* rs2049046 and rs11030094 to increase the response of antidepressant treatment in MDD patients [82].

## 5. Major Risk Factors of MDD

### 5.1. Environmental Factors

There are many factors that are likely responsible for the development of MDD, and of all these, life experiences are most studied. Risk factors related to life events and changes include but are not limited to natural calamities, financial constraints, bullying, social seclusion, social stress, diagnosis of a serious medical disorder, loss of loved ones, and childbirth [15]. The prevailing COVID-19 pandemic and the resultant restrictions have greatly affected mental health. A study conducted by Evans et al. (2021) reported a significant rise in the symptoms of MDD during the COVID-19 pandemic. Alcohol use may be a contributory factor to this [83]. The use of certain medications such as alpha-interferon, isotretinoin, and rimonabant also resulted in a higher risk of developing MDD [84,85,86].

### 5.2. Genetic Factors and Heritability

MDD is a multifactorial disorder caused not only by the environment but by genes as well, and this is supported by genetic and twin studies [87]. Heritability is a key concept in genetics that estimates how much of the variance in a trait is due to genetic factors, or, put another way, how much resemblance exists between parents and offspring. Heritability is typically estimated from twin studies based on the expected genetics and environmental sharing within the twin pairs. Heritability is population and time-specific [88]. The heritability of psychiatric disorders ranges from moderately to highly heritable, but the degree to which genetic variation is exclusive to individual disorders or shared among them is unclear. Twin studies and other similar studies suggest a moderately heritable component (37%) for MDD, but it may range between 26% and 49% [89]. Overall, heritability estimates from twin studies are fairly similar, but twin studies rely on some key assumptions, particularly the common environment assumptions between monozygotic and dizygotic twin pairs. This was relevant to the missing heritability reported by early genome-wide association studies (GWAS), which led some studies to suggest that the disease’s twin heritability might have been overestimated [90]. However, there is a growing literature on heritability estimation based on population-based registers or electronic health record (EHR) data. These studies reconstruct the extended pedigree based on either recorded or inferred familial relationships, and the estimates here are fairly similar to those from twin studies [91,92,93]. Heritability is significantly higher in females than in males. Most, but not all, genetic risk is shared across sexes. Given the moderate heritability of MDD, it would be useful to identify clinical subtypes that are more heritable. One successful example was the ascertainment of recurrent MDD cases in women to enhance the GWAS power [94]. Early evidence established recurrent MDD as a more heritable form of the disorder, but recent studies using larger population-based data were able to extend the evidence to suggest that disorder subtypes based on the early age of onset, comorbid anxiety disorder, higher severity of MDD and postpartum depression are more heritable forms of MDD [95].

The combination of high prevalence and low heritability has made it challenging to map genes for MDD. There has also been a debate about the genetic architecture of MDD. The common-disease/common-variant model posits that psychiatric disorders are caused by the combined effect of many common variants, each of small effect. The multiple rare variant model argues that psychiatric disorders are caused by large-effect mutations in single genes, with each case having different casual mutations. The results from genetic studies described later indicate that both models are relevant to a complex polygenic disorder such as MDD, where contributory genetic variants exist along the continua of allele frequency (from rare to common) and effect size (from small to large), with the exception of common large-effect genetic variants due to the pressures of selection. This genetic complexity requires a range of study designs to find the susceptibility genes for MDD.

## 6. Poor Replication of MDD-Associated Genes Identified by Candidate Gene Studies

Early research on the identification of genes for MDD focused mainly on particular candidate polymorphisms in genes hypothesized to cause depression [24]. These candidate genes were selected on the basis of their role in disordered physiological processes and drug mechanisms related to the known biological basis of the disorder [79]. More than 1500 studies have been reported that studied variants in more than 200 candidate genes with conflicting results. Studies were largely underpowered with insufficient sample sizes to detect associations with variants of low effect, resulting in inconsistent findings and failure to achieve independent replications. This intensified the need for alternative methods to the candidate gene approach for reliable, replicable genetic findings [95]. In fact, a high-level National Institute of Mental Health genomics panel recommended, “Candidate gene studies of psychopathologic, cognitive or behavioral phenotypes should be abandoned in favor of well-powered, unbiased association studies” [96]. Therefore, the candidate gene approach has been replaced with more robust GWAS that leverages large sample sizes, advanced genomics technologies such as genotyping arrays and next-generation sequencing (NGS), imputation and reference panels to provide genome-wide coverage of common SNPs and a range of biostatistical methods to detect novel and replicable genetic associations for MDD.

## 7. Contribution of GWAS to Our Understanding of the Genetic Architecture of MDD

GWAS test for differences in the allelic frequency of genetic variants between groups of individuals who share similar ancestry but are different phenotypically (e.g., MDD cases vs. controls). Single nucleotide polymorphisms (SNPs) are the most widely studied genetic variants in GWAS using genotyping arrays, although these arrays can also detect large copy number variants (CNVs) [96,97]. GWAS identify SNPs that have statistically significant associations with phenotypes of interest based on the association reaching genome-wide significance (based on a correction for all SNPs tested) and showing evidence of independent replication. Over the last 15 years, GWAS have led to some remarkable discoveries in the field of human genetics. They have successfully been applied to understand disease biology, estimate heritabilities, calculate genetic correlations between phenotypes, develop risk prediction using genetic variables, appraise drug development programmes, and draw inferences about the potential causal relationship between risk factors and health outcomes [98]. In psychiatry, the findings of GWAS for MDD and associated traits have proven to be more productive than earlier linkage studies and candidate gene association studies [96]. Hundreds of loci have now been reported in different large-sample GWAS that are significantly and strongly associated with MDD and associated traits [95,99].

Early GWAS by the initial Psychiatric Genomics Consortium (PGC) MDD mega-analysis (9240 cases; [100]) or by the CHARGE meta-analysis of depressive symptoms (*n* = 34,549; [101]) struggled to detect any significant results. However, as sample sizes increased, numbers of genome-wide significant loci began to increase. The estimates of the proportion of variance attributable to common SNPs (SNP heritability, h^2^_SNP_) remained relatively constant even after sample sizes with >100,000 cases were used (Figure 1). A meta-analysis of depressive symptoms identified 2 loci (161,460 individuals; [17]), and an analysis of self-reported major depression identified 15 loci in a sample containing 75,607 cases and 231,747 controls [102]. Studies since then have identified 14 loci (113,769 cases and controls 208,811 [103]), 44 loci (135,458 cases and 344,901 controls; [5]), 102 loci (246,363 cases and 561,190 controls; [4]) and 178 loci (366,434 cases and 847,433; [99]) with continually increasing sample sizes.

The largest study identifying 178 loci was a bi-ancestral meta-analysis using individuals of European ancestry and African ancestry from the Million Veteran Program (MVP), 23andMe, UK Biobank and FinnGen and included many samples used in previous MDD GWAS [99]. Similar to those GWAS, this study used different depression phenotypes to maximize the case sample available for analysis. For MVP, different diagnostic definitions were available: an ICD–based phenotype using EHR data, self-reported physician diagnosis of depression and the two-item Patient Health Questionnaire scale of depressive symptoms in the past 2 weeks, included in the MVP baseline survey. Genetic correlations among these traits were high. Of these, the EHR-derived ICD codes for MDD were chosen for the primary meta-analysis because they had the highest heritability and largest sample size. The genetic correlations between MVP, 23andMe, UK Biobank and FinnGen were high (>0.71) despite these studies using a variety of phenotype definitions, e.g., structured interview-based clinical diagnosis of MDD, self-reported treatment or self-reported diagnosis. The consistency in collection for the large UKB and MVP samples reduced ascertainment heterogeneity within samples and likely increased study power. The genetic findings were replicated in an entirely independent sample of 1.3 million participants from 23andMe, where cases of depression were determined by participants’ answers to questions about having been diagnosed or treated for depression. These results demonstrated the consistency of GWAS findings once adequate power is achieved and the utility of different phenotyping methods even though they may reduce the specificity of findings for a core MDD phenotype [99].

This largest GWAS of MDD confirmed many previously reported loci and identified new ones. In addition to depression-related phenotypes, the most significant genetic correlation for MDD was tiredness/lethargy. The most significantly genetically correlated brain imaging phenotype was left subcallosal cortex gray matter volume. A tissue-based transcriptome-wide association study (TWAS) identified 153 genes in 14 tissues (13 brain regions plus whole blood). A follow-up variant prioritization that tested if the same locus is shared between GWAS and tissue-specific expression quantitative trait loci (eQTL) identified five gene–tissue pairs: *CCDC71*–amygdala, *FADS1*–cerebellar hemisphere, *SPPL3*–frontal cortex, *TRAF3*–hypothalamus and *LAMB2*–whole blood. The protein product of *CCDC71* and *FADS1* is involved in the regulation of fatty acids and lipid metabolism. Several studies have been reported for increased risk of MDD due to depletion of omega-3 fatty acid, although the mechanism of how omega-3 supplementation treats depression is still unclear [99,104]. The protein product of *SPPL3* (signal peptide peptidase-like 3) is involved in various cellular signaling cascades, including the T cell receptor signalling pathway and positive regulation of protein dephosphorylation [105]. The protein product of *TRAF3* (TNF receptor-associated factor 3) controls the response of type-1 interferon, where studies have demonstrated an increased risk of MDD development associated with treatments based on interferon [106]. The protein product of *LAMB2* (laminin subunit beta 2) is an integral part of neuromuscular junctions and is involved in neuropathic pains, a disturbance that influences the brain pathways in MDD [107]. Another top finding from the TWAS analysis was *DRD2* (D2 dopamine receptor) with a significantly predicted decreased expression in the nucleus accumbens, which is a drug target and critical part of the mesolimbic dopamine reward circuit that has long been implicated in depression. The authors noted this finding from hypothesis-free investigations as “remarkable with respect to known biology” and provided optimism that novel genetic findings could help elucidate molecular mechanisms and highlight new therapeutic targets. Overall tissue expression analysis identified enrichment across all brain tissues and the pituitary, with the strongest findings for the frontal cortex Brodmann area (BA) 9, cortex, cerebellar hemisphere, anterior cingulate cortex, BA24 and cerebellum, and no enrichment in non-neuronal tissue. The most enriched biological processes were involved in nervous system development, synapse assembly, and organization. Gene-drug relationships were explored by drug mapping and identified four drugs that are either estrogen receptor agonists (diethylstilbestrol, implanon (etonogestrel implant)) or anti-estrogens (tamoxifen and raloxifene), suggesting opportunities for re-purposing, plus nicotine, cocaine, cyclothiazide, felbamate and riluzole. The latter three drugs have been shown to modulate glutamatergic activity, an interesting finding in the context of glutamate’s emerging role in mood disorders [99].

## 8. GWAS Identify Genetic Correlations between MDD and Other Disorders and Traits

GWAS have revealed the polygenic and pleiotropic nature of psychiatric disorders. There exists a high level of psychiatric comorbidity, and the heritability of psychiatric disorders based on twin studies ranges from ~40 to 80% [1,108,109]. Many casual variants are shared across these disorders and traits, as pleiotropy is pervasive [110,111]. Shared genetics between phenotypes can be calculated by genetic correlation, which is estimated from GWAS summary statistics using linkage disequilibrium score regression (LDSR). Initial GWAS based on large sample sizes highlighted that the strongest significant genetic correlations for MDD were with other major adult psychiatric disorders (schizophrenia and bipolar disorder (BD)), neuroticism, college completion and educational attainment, measures of sleep quality, coronary artery disease, triglycerides, body fat, waist-to-hip ratio, an earlier age of smoking initiation, and traits related to female reproductive life events (age at menarche and age at menopause; [4,5]). The most recent and largest GWAS reports significant genetic correlations for MDD with 668 diseases, disorders and traits from most categories of biobank-type phenotypes tested [99].

## 9. Combined Analysis of MDD with Brain-Related Phenotypes Increases Discovery of New Loci

Based on genetic correlations, some phenotypes have been co-analyzed with MDD to identify additional risk loci. A study of the genetics of the mood disorder spectrum meta-analyzing MDD and bipolar disorder GWAS data identified 73 genome-wide significant loci, 15 of which were novel for mood disorders. Genetic correlations revealed that type 2 bipolar disorder correlates strongly with recurrent and single episode major depressive disorder [112]. Interestingly, post-GWAS analyses identified both similarities and differences between the mood disorders. For example, cell types enriched for MDD-associated genes were also enriched for BD-associated genes (dopaminergic neuroblasts; dopaminergic, GABAergic and midbrain nucleus neurons from embryonic mice; interneurons; and medium spiny neurons), but some cell types enriched for BD-associated genes had no enrichment for MDD-associated genes (pyramidal cells from the CA1 region of the hippocampus and the somatosensory cortex, and striatal interneurons; [112].

Anxiety and stress-related disorders (ASRDs) have overlapping symptoms and high rates of comorbidity with MDD. There is a positive and significant genetic correlation between these disorders. The genetic correlation between the risk of anxiety disorder with major depressive disorder was found to be positive, at 0.83 ± 0.16, *p* = 1.97 × 10^–7^ [113]. A cross-trait meta-analysis identified 5 pleiotropic loci simultaneously associated with MDD and ASRD, and highlighted *NUP210L* as a potential mediator of the genetic correlation between these disorders (GNOVA: *rho* = 0.59, *se* = 0.01, *p* = 5.32 × 10^–45^) [114]. Cognitive impairment is a feature of MDD. Although there is a non-significant genetic correlation between general intelligence and MDD, GWAS analysis identified 92 shared loci between these two phenotypes, with 69 and 64 loci novel for MD and intelligence, respectively [115]. Here, there was a balanced mixture of directional effects among the loci shared between MDD and general intelligence, meaning that SNP alleles associated with increased risk of MDD are associated with lower intelligence at some loci and higher intelligence at other loci.

Late-onset Alzheimer’s disease (LOAD) exhibits comorbidity with MDD and anxiety, and MDD is associated with an increased risk of LOAD. Pleiotropy analysis found moderate enrichment for SNPs associated with LOAD across increasingly stringent levels of significance with the MDD GWAS association. Associated SNPs mapped to 40 genes, including 9 genes on chromosome 11 in the known LOAD-risk regions that contain the *SPI1* gene and *MS4A* genes cluster, plus other novel pleiotropic risk-loci for LOAD conditional with MDD [116]. Attention deficit hyperactivity disorder (ADHD) in childhood or adolescence is associated with an increased risk of subsequent depression, and GWAS detects a genetic correlation between the two disorders (*r_g_* = 0.52 (0.04)). Of 14 linkage disequilibrium-independent SNPs associated with ADHD and MDD separately and in a cross disorder meta-analysis (at genome-wide significance levels), 9 were novel findings for these disorders and revealed a molecular genetic basis for the overlap between disorders [117]. Cross-trait meta-analyses identified 89 genomic loci as being shared between MDD and insomnia, indicating a substantial shared genetic liability between these 2 interconnected conditions [118].

## 10. Exploring Causal Relationships

Genetic correlations between MDD and other disorders and traits may reflect the pleiotropic effects of genes and biological influences across both phenotypes but also may be a causal effect of MDD on other traits (for example, depression influencing sleep quality) or the causal effect of other phenotypes on MDD (for example, sleep quality leading to depression). Mendelian randomisation (MR) estimates causal relationships between genetically based traits by taking SNPs associated with one trait, the exposure, and using them as genetic instruments to test for causal effects of the exposure on a second trait, the outcome [119]. MR has identified a putative causal effect of body mass index, years of education and interleukin (IL)-6 levels on MDD, a putative causal effect of MDD on a smoking phenotype (ever versus never smoked), metabolic syndrome and its components (waist circumference, hypertension, triglycerides, high-density lipoprotein cholesterol), coronary artery disease, stroke, and inflammatory bowel disease, and bidirectional causal effects for MDD each of the following: schizophrenia, neuroticism, pain, osteoarthritis pain, insomnia and atopic diseases (asthma, hay fever, and eczema) [4,5,118,120,121,122,123,124,125,126]. The latter bidirectional results are consistent with a shared biological basis for MDD and these disorders and traits. Overall, these MR findings highlight different phenotypes as potentially targetable risk factors for the treatment of MDD and opportunities to reduce comorbidities if MDD can be successfully treated or prevented.

## 11. Rare Variant Analysis of MDD

An analysis of 488,366 participants from the UK Biobank identified 3 rare large (>100 kilobases (kb)) CNVs (1q21.1 duplication, Prader–Willi syndrome duplication, and 16p11.2 duplication) that were robustly associated with self-reported depression [127]. These CNVs have previously been implicated in neurodevelopmental disorders such as autism, intellectual disability and schizophrenia, again supporting shared etiology between MDD and these disorders. Patients with MDD have also been reported to carry significantly more short deletions (CNVs < 100 kb) than control subjects. These are primarily in intergenic regions, and no single CNV was associated at genome-wide significant levels [128].

NGS-based sequencing applications such as whole-exome sequencing (WES) and whole-genome sequencing (WGS) allow for the detection of rare and low-frequency sequence variants that may affect the risk of MDD [98,129]. A low-coverage WGS study of 5303 Chinese women with recurrent MDD (selected to reduce phenotypic heterogeneity) and 5337 controls by the CONVERGE consortium identified 2 risk variants: 1 near the *SIRT1* gene and another in an intron of the *LHPP* gene [94]. Small-scale WES studies have not identified robustly associated loci for MDD [130,131]. Neither has the largest WES study to date using 22,886 cases and 176,486 controls from UK Biobank, but the mood disorder phenotype was based on having seen a psychiatrist for “nerves, anxiety, tension or depression” [132]. As was the case for GWAS, several iterations of study designs with an ever-increasing sample size will most likely be needed to find robustly associated rare sequence variants for MDD. The reward will be a more direct identification of risk genes based on the locations of causative variants, particularly if detected within exonic regions.

## 12. Challenges for Genetic Studies of MDD

In comparison to schizophrenia, where the critical inflection point for GWAS lay at 15,000 cases, after which the discovery of a new locus increased linearly at an average of 230 new cases per new locus [133], each new genome-wide significant locus for MDD takes considerably more effort. The inflection point for MDD was found at 75,000–100,000 cases, with the discovery of a new locus then increasing linearly at an average of 1500 new cases per new locus, reflecting the disorder’s lower heritability and higher prevalence in the population [95,134]. As larger samples have succeeded in finding new loci, the h^2^_SNP_ from MDD GWAS that has had >100,000 cases remained in a range of 0.087–0.113 (Figure 1), which is less than one-third of the *h*^2^ estimated from twin or family studies [135]. The missing heritability may be explained by even larger GWAS, the discovery of rare variants via NGS methods, or it may be that twin and family studies have overestimated heritability [136].

The findings obtained from GWAS analysis can be used to estimate individual-level risk measures by estimating polygenic risk scores (PRSs) that attempt to predict the genetic liability of MDD in a set of individuals independent of the original GWAS. PRS aggregates the number of risk alleles an individual carries by weighting each variant by its effect size, thereby summarizing the large genotype matrix into a single variable per individual [96]. Nagelkerke’s *R*^2^ is used to calculate the proportion of phenotypic variance explained on the liability scale using PRSs. Despite a large number of loci identified, the variance prediction capability of MDD is ultra-low (Nagelkerke’s *R*^2^ of 1.5–3.2% on the liability scale [4]) in comparison to other psychiatric disorders such as schizophrenia (Nagelkerke’s *R*^2^ of 7.7% on liability scale [137]). Therefore, MDD-based PRSs are not going to have any immediate clinical utility as a diagnostic tool. However, PRSs may have uses post-diagnosis of a mood disorder, for example, in discrimination between BD and MDD cases [138] and for treatment stratification [139].

A major limitation of PRSs is that they are highly sensitive to ethnic background and thus have the greatest predictive power where the discovery sample used for GWAS and the independent target sample have the same genetic ancestry [140]. GWAS has a major diversity problem, as most have used samples of European descent [141]. GWAS in more diverse samples is emerging, such as a study of 15,771 individuals with depression and 178,777 control participants of East Asian ancestry [142]. Here, several depression loci were not transferable between studies of cohorts of East Asian and European ancestry. Transancestry genetic correlations between the depression outcomes in these cohorts were moderate (max *r* = 0.558) and considerably lower than those reported for schizophrenia (*r*  =  0.98) [142]. Therefore, genetic findings for MDD may not transfer beyond the studied population. More diverse GWAS can allow for more accurate PRS profiling and help the fine mapping of associated loci based on differential LD patterns in different populations.

Many studies have sought to explore gene-environment interactions through investigations of known major environmental factors for MDD, such as stressful life events (e.g., childhood trauma). Results here have been inconsistent, with some data indicating that a higher h^2^_SNP_ for MDD is evident among individuals reporting lifetime trauma exposure [143], whereas other data suggests that in the absence of environmental adversity, MDD may have a stronger genetic basis [112]. Using measures of early-life risk factors and sociodemographic variables from UK Biobank, gene-environment interactions were reported for childhood trauma and the Townsend Index of material deprivation, where the effect of depression-based PRS was enhanced in participants exposed to more adverse social/socioeconomic environments [144]. The study reported no significant interactions between the depression-based PRS and adulthood trauma, recent stressful life events and household income. As genetic studies progress and further biobanks come on-stream, ideally with greater longitudinal data, the scope for gene-environment studies of MDD will increase.

## 13. Summary

As predicted [133], GWAS of MDD have been successful despite the obstacles of moderate heritability, high prevalence, heterogeneity of genetic and non-genetic factors, and heterogeneity of samples. We now have a working knowledge, although incomplete, about many of the biological processes, cell types, tissues and druggable targets that are enriched for MDD risk genes. We are starting to understand the genetic relationship between MDD and many psychiatric and non-psychiatric disorders and traits. We are uncovering the complexity of MDD genetics in diverse populations. We are starting to test new genetic findings within the paradigm of gene-environment interactions. All this provides a foundation for the re-evaluation of previous hypotheses and the development and testing of new hypotheses about the molecular mechanisms for this multi-factorial disorder. Although a comprehensive guiding strategy for the application of genetics in the clinical management of MDD is still many years away, we can be confident that accelerating genetics and functional genomics research can contribute to precision medicine for this very common and hugely disabling disorder.

## Figures and Tables

**Figure 1 genes-13-00646-f001:**
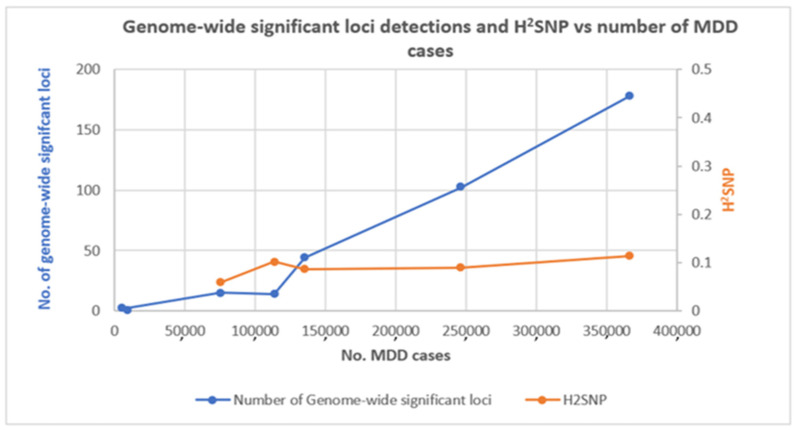
A plot of the number of genome-wide significant loci (left-hand-side *x*-axis and blue line) and h^2^_SNP_ (right-hand-side *x*-axis and orange line) from GWAS of MDD showing increases in the number of significant findings with increasing the sample sizes. From left to right, the dots represent the following studies: PGC (2013) [100], CONVERGE Consortium (2015) [94], Hyde et al. (2016) [102], Howard et al. (2018) [103], Wray et al. (2018) [104], Howard et al. (2019) [4] and Levey at al. (2021) [99].

**Table 1 genes-13-00646-t001:** US FDA-approved antidepressants used in the management of major depressive disorder (MDD).

Pharmacological Class	Drugs
Selective seretonin reuptake inhibitors (SSRIs)	Citalopram, escitalopram, fluoxetine, fluvoxamine, sertraline and paroxetine
Serotonin and norepinephrine reuptake in-hibitors (SNRIs)	Duloxetine, desvenlafaxine and venlafaxine
Monoamine oxidase inhibitors (MAOIs)	Isocarboxazid, phenelzine, selegiline and tranylcypromine
Tricyclic antidepressants (TCAs)	Amitriptyline, desipramine, doxepine, maprotiline, nortriptyline, protriptyline and imipramine
Noradrenergic and specific serotonergic modulators	Mirtazapine
Norepinephrine-dopamine reuptake inhibitors	Bupropion
Multimodal antidepressants	Vortioxetine
Serotonin modulators	Trazodone and nefazodone
MT1/MT2 agonists and 5-HT2C antagonists	Agomelatine
Serotonin reuptake inhibitors and 5-HT1A-receptor partial agonists	Vilazodone
Neurosteroids	Bresanolone
Newer agents like NMDA receptor antagonists	Esketamine

## Data Availability

Data were directly downloaded from published studies and all additional generated data is contained within this manuscript.

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
