# Peer review of "Major Depressive Disorder: Existing Hypotheses about Pathophysiological Mechanisms and New Genetic Findings"

_genes, 2022, doi:10.3390/genes13040646_

Round 1

Reviewer 1 Report

The manuscript “Major Depressive Disorder: existing hypotheses about pathophysiological mechanisms and new genetic findings” is review of the MDD literature with a focus on existing hypotheses about the pathophysiological mechanisms and GWAS loci. It is well-written and provides a fair presentation of the existing literature. I have only minor comments for the authors:

  1. The authors did not mention that psychosis can be a clinical feature of MDD. Including psychosis as part of the potential clinical manifestation of MDD has educational value as it seems many of the readers of this journal may not know this important fact.
  2. There are many existing hypotheses about the pathophysiological mechanisms of MDD and it is unrealistic to expect the authors to include all of them, however, the manuscript would be strengthened by inclusion of the neuroplasticity/neurogenesis hypothesis given the evidence supporting it is significant and its ability to explain multiple clinical and neuroimaging features of MDD.
  3. The identification of GWAS loci associated with MDD is exciting, however, one the next major challenges of the field will be to understand how the variants associated with MDD affect biology and thus contribute to risk for MDD. It would strengthen the manuscript if the authors provided a brief discussion of how researchers might elucidate the biological function of MDD-associated variants and the mechanisms by which they contribute to MDD risk.

Author Response

Reviewer 1

Issue 1; The authors did not mention that psychosis can be a clinical feature of MDD. Including psychosis as part of the potential clinical manifestation of MDD has educational value as it seems many of the readers of this journal may not know this important fact.

Response 1; Psychosis is added as clinical manifestation in section 2 of MS lines 60, 61, and 63 under section 2. Diagnosis and MDD phenotypes.

Issue 2; There are many existing hypotheses about the pathophysiological mechanisms of MDD and it is unrealistic to expect the authors to include all of them, however, the manuscript would be strengthened by inclusion of the neuroplasticity/neurogenesis hypothesis given the evidence supporting it is significant and its ability to explain multiple clinical and neuroimaging features of MDD.

Response 2; The neuroplasticity/neurogenesis hypothesis is added as section 4.6. in MS from line 282-298.

Issue 3; The identification of GWAS loci associated with MDD is exciting, however, one the next major challenges of the field will be to understand how the variants associated with MDD affect biology and thus contribute to risk for MDD. It would strengthen the manuscript if the authors provided a brief discussion of how researchers might elucidate the biological function of MDD-associated variants and the mechanisms by which they contribute to MDD risk.

Response 3; The biological function of associated genes is mentioned in section 4, however, a brief discussion on biological role of genetic variants mentioned in section 7 line 431-442 has been added.

Reviewer 2 Report

The goal of the authors is to provide a bird's eye view of recent genetic studies in MDD, the goal of which, in part, is to reveal biological basis of this debilitating condition.  However, often they abbreviated their discussions of important points, to the extent that the reader is left under informed or confused or both. First, ascertainment is a critical issue clinically and for research purposes, including genetics analysis. The authors' discussion of this important issue in the beginning of their piece was superficial, and they only returned to it in their discussion of the Levey et al Nature Neuroscience paper. In that work, Levey et al were very careful to employ several distinct strategies in their attempts to mitigate its potentially negative effects. It would have been very helpful if Kamran et al would have framed their initial discussion of ascertainment from the perspective of the ways it might be taken into consideration, as a way to help the reader judge the quality of the studies which they described. This was especially true in the section in which they discussed various contributions, including small sample sizes, to poor replication (Section 6).

As another example, the authors seem to take for granted the complexity of the SLC6A4 genetic region, as they highlight that 'hits' in the region might be in or near mir-16 binding sites. Not only is this problematic in its own right, but the authors missed an opportunity to explain more generally how structural variation within a genomic region may contribute to the association of the region with disease phenotype. 

Additional issues to consider:

  1. Providing a more thorough discussion of how epigenetic alterations might be associated with MDD would be helpful, especially in the context of potential environmental contributory factors.
  2. In the figure, the heritability appears to increased between 75,000 and 100,000 cases, but remains constant thereafter. In the immediate discussion of the figure the authors seem to suggest that it continues to rise, and only in line ~553 do they indicate is remains roughly constant in regions >100,000 cases. 
  3. In line 399 the authors mention "in the latter study" - presumably the Levey et al study is meant, but you could be more explicit.
  4. To what extent are correlations of the findings for MDD with other brain-related phenotypes evidence of underlying biological connections, and not imprecise or overlapping ascertainments of the various conditions? 
  5. In line 425 authors quote Levey et al (using italics for emphasis) that their hypothesis-free results are remarkable with respect to known biology. Disappointingly, aside from the drug relations and the eQTL gene-tissue pairs, which are not unique to MDD, they don't describe those relations in any detail. This is a major shortcoming. 

Author Response

Reviewer 2

Issue 1; The goal of the authors is to provide a bird's eye view of recent genetic studies in MDD, the goal of which, in part, is to reveal biological basis of this debilitating condition. However, often they abbreviated their discussions of important points, to the extent that the reader is left under informed or confused or both. First, ascertainment is a critical issue clinically and for research purposes, including genetics analysis. The authors' discussion of this important issue in the beginning of their piece was superficial, and they only returned to it in their discussion of the Levey et al Nature Neuroscience paper. In that work, Levey et al were very careful to employ several distinct strategies in their attempts to mitigate its potentially negative effects. It would have been very helpful if Kamran et al would have framed their initial discussion of ascertainment from the perspective of the ways it might be taken into consideration, as a way to help the reader judge the quality of the studies which they described. This was especially true in the section in which they discussed various contributions, including small sample sizes, to poor replication (Section 6).

Response 1; This is an important issue. Firstly, we would argue that differences in ascertainment methods was not a major contributor to the failure of candidate gene studies (discussed in section 6). Here, the major shortcoming was sample size and lack of statistical power to detect common variants of small effect. This was also the case for other common psychiatric disorders that have much higher heritability than MDD such as schizophrenia. There, small samples, although only including cases that were ascertained following careful recruitment using research diagnostic tools, consistently failed to produce results that replicated across independent samples. Interesting, in recent large schizophrenia GWAS, analysis shows that cases recruited using a research diagnosis and cases recruited just by clinical assessment are not genetically different from each other, highlighting that different ascertainment strategies can aid gene discovery by facilitating larger sample collection. This relates back to our discussion of this issue of multiple ascertainment methods in this review.

To address the reviewer’s point, we have just added a sentence (line 72) to the initial discussion of diagnosis and ascertainment to indicate that we will expand on this issue in section 7 on GWAS. We believe that it is better to cover this issue after the concept of GWAS and the application of GWAS to MDD has been described, to give it greater context. In section 7, taking on board the reviewer’s helpful suggestion, we have included extra text on the efforts of  Levey et al to employ several distinct strategies in their attempts to mitigate the potentially negative effects of ascertainment biases. These analyses show that the genetic basis of these different phenotypic definitions are highly correlated with each other, line 409-416

Issue 2; The authors seem to take for granted the complexity of the SLC6A4 genetic region, as they highlight that 'hits' in the region might be in or near mir-16 binding sites. Not only is this problematic in its own right, but the authors missed an opportunity to explain more generally how structural variation within a genomic region may contribute to the association of the region with disease phenotype.

Response 2; Structural variation in genomic region of SLC6A4 gene and their effect on expression level is added in more detail in section 4.2 line 177-178, 180-185

Issue 3; Providing a more thorough discussion of how epigenetic alterations might be associated with MDD would be helpful, especially in the context of potential environmental contributory factors.

Response 3; A thorough discussion of the most investigated epigenetic changes in depression as a potential environmental contributory factor is added in section 4.2 line 199-207.

Issue 4; In the figure, the heritability appears to increase between 75,000 and 100,000 cases, but remains constant thereafter. In the immediate discussion of the figure the authors seem to suggest that it continues to rise, and only in line ~553 do they indicate is remains roughly constant in regions >100,000 cases.

Response 4; Corrections are made in section 7 line 389-391.

Issue 5; In line 399 the authors mention "in the latter study" - presumably the Levey et al study is meant, but you could be more explicit.

Response 5: Corrections are made to address this point in line 402.

Issue 6; To what extent are correlations of the findings for MDD with other brain-related phenotypes evidence of underlying biological connections, and not imprecise or overlapping ascertainments of the various conditions?

Response 6; Further detail on genetic correlations is added in section 9 line 480-482, 491-492, 495, and 511.

Issue 7; In line 425 authors quote Levey et al (using italics for emphasis) that their hypothesis-free results are remarkable with respect to known biology. Disappointingly, aside from the drug relations and the eQTL gene-tissue pairs, which are not unique to MDD, they don't describe those relations in any detail. This is a major shortcoming.

Response 7; The Levey et al. study is one of many GWAS that we have included for review here. It is arguably the most important given that it is the most recent, has the largest sample and identifies the most genome-wide significant loci. However, given the volume of results reported in that study, we can only highlight some of the most important findings here and direct readers to investigate this study themselves. We do devote two full paragraphs and >680 words to this study alone. To address the reviewer’s concern, we have now included additional text on five genes that Levey et al. highlight based on their efforts to prioritize variants using biologically and statistically informed annotations. These genes and their target tissues were identified by integrating both transcriptomics and CADD score prioritized variants. This method aided in the identification of shared causal loci for phenotype and tissue-specific eQTLs, line 424-435.

Round 2

Reviewer 2 Report

In my mind, this paper needs to be re-written from the ground up. It is disorganized in thought and presentation. A few issues for me:

The authors point out that there are no biomarkers that can be used to confirm diagnosis of MDD for either treatment or research purposes (lines 46 – 47). But a few lines later they mention several biomarkers that are being investigated to optimize treatment, including electrophysiology, neuroimaging, and peripheral proteomics. There are also metabolomics and blood-level transcriptomics studies. The authors should either avoid blanket statements or devote a section to these studies. They might dovetail with the GWA studies.

The authors elaborated on their discussion of environmental influences and epigenetics, but that section (lines 199 – 207) seems misplaced. At the very least it should be moved up, since the authors discuss DNA methylation earlier, in the context of the serotonin receptor.

In their discussion of miR-16 regulation, I’m not sure I understand to what the phrase ‘this gene’ in line 183 refers.

In their discussion of the corticoid receptors, to what does ‘in the cytosolic region’ refer (line 193)?

The authors improved their discussion around my main concern, that ascertainment is a critical issue for reliable GWA studies of MDD. However:

  1. They assert that the low success rate of treating MDD patients reflects the fact that diagnosis is solely dependent on behavioral systems, which can vary from patient to patient and even temporally within patients. In other words, treatment fails because of poor diagnostics, but this does not affect GWA studies?
  2. After stressing the sufficiency of methods currently used in large scale and meta-analysis, the authors then (lines 568-569) used the ‘range of methods to identify cases’ (which, in my mind, encompasses ascertainment) as part of their explanation for the high ratio of cases to newly identified loci, compared with other conditions like schizophrenia.

Author Response

In my mind, this paper needs to be re-written from the ground up. It is disorganized in thought and presentation. A few issues for me:

The authors point out that there are no biomarkers that can be used to confirm diagnosis of MDD for either treatment or research purposes (lines 46 – 47). But a few lines later they mention several biomarkers that are being investigated to optimize treatment, including electrophysiology, neuroimaging, and peripheral proteomics. There are also metabolomics and blood-level transcriptomics studies. The authors should either avoid blanket statements or devote a section to these studies. They might dovetail with the GWA studies.

R: Thanks for highlighting this issue. Biomarkers was not intended to be a major focus of the review and with the review’s length already, we think it best not to include a section on those studies. The point we have made in lines 46-47 is that “there are no specific biomarkers that can be used to confirm the diagnosis of MDD”. But later in line 91-92 we highlight that “Individual biomarkers and clinical characteristics are used to predict the efficiency of management strategies”, not diagnosis. Therefore, we don’t believe that we are contradicting ourselves. However, we have taken the reviewer’s comment on board and to avoid confusion, we have modified lines 91-92 to read “Although not used for diagnosis, individual biomarkers and clinical characteristics are used to predict the efficiency of management strategies.”

The authors elaborated on their discussion of environmental influences and epigenetics, but that section (lines 199 – 207) seems misplaced. At the very least it should be moved up, since the authors discuss DNA methylation earlier, in the context of the serotonin receptor.

R: We have taken the reviewer’s suggestion and moved this text on environmental influences and epigenetics from the section on Glucocorticoids to the section on Serotonin.

In their discussion of miR-16 regulation, I’m not sure I understand to what the phrase ‘this gene’ in line 183 refers.

R: That was a reference to 5-HT – we have made an edit to clarify that.

In their discussion of the corticoid receptors, to what does ‘in the cytosolic region’ refer (line 193)?

R: That was a typographical error and has now been removed.

The authors improved their discussion around my main concern, that ascertainment is a critical issue for reliable GWA studies of MDD. However:

  1. They assert that the low success rate of treating MDD patients reflects the fact that diagnosis is solely dependent on behavioral systems, which can vary from patient to patient and even temporally within patients. In other words, treatment fails because of poor diagnostics, but this does not affect GWA studies?
  2. After stressing the sufficiency of methods currently used in large scale and meta-analysis, the authors then (lines 568-569) used the ‘range of methods to identify cases’ (which, in my mind, encompasses ascertainment) as part of their explanation for the high ratio of cases to newly identified loci, compared with other conditions like schizophrenia.

R: Yes, treatment may fail due to the heterogeneity of the disorder (which also makes  it difficult to diagnose) and heterogeneity has affected the power of GWAS. However, the genetic correlations between the different samples in the Levey et al. (2021) GWAS were high (>0.71) despite these studies using a variety of phenotype definitions, e.g. structured inter-view-based clinical diagnosis of MDD, self-reported treatment or self-reported diagnosis. In relation to our reference to the “range of methods to identify cases” as a contributor to the high ratio of cases to newly identified loci, yes, we may appear here to be contradicting ourselves. We have edited that section to say that the likely reasons for this high ratio of cases to newly identified loci are “the disorder’s lower heritability and higher prevalence in the population”.

Round 3

Reviewer 2 Report

I thank the authors for taking my comments into consideration.